# Muonic lithium atoms:
# Nuclear structure corrections to the Lamb shift

**Simone Salvatore Li Muli[1]⋆, Anna Poggialini[2] and Sonia Bacca[1]†**

**1** Institut für Kernphysik and PRISMA+ Cluster of Excellence,
Johannes Gutenberg-Universität, Mainz, Germany
**2** Università degli Studi di Siena, Facoltà di Fisica e Tecnologie Avanzate, Siena, Italy

⋆ silimuli@uni-mainz.de † s.bacca@uni-mainz.de

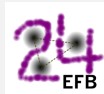

*Proceedings for the 24th edition of European Few Body Conference,
Surrey, UK, 2-6 September 2019*

## Abstract

In view of the future plans to measure the Lamb shift in muonic Lithium atoms we address the microscopic theory of the $\mu$-$^6$Li$^{2+}$ and $\mu$-$^7$Li$^{2+}$ systems. The goal of the CREMA collaboration is to measure the Lamb shift to extract the charge radius with high precision and compare it to electron scattering data or atomic spectroscopy to see if interesting puzzles, such as the proton and deuteron radius puzzles, arise. For this experiment to be successful, theoretical information on the nuclear structure corrections to the Lamb shift is needed. For $\mu$-$^6$Li$^{2+}$ and $\mu$-$^7$Li$^{2+}$ there exist only estimates of nuclear structure corrections based on experimental data that suffer from very large uncertainties. We present the first steps towards an ab initio computation of these quantities using few-body techniques.

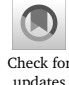

# 1 Introduction

Since the discovery of the "proton radius puzzle", light muonic atoms have attracted a lot of attention. Hydrogen-like systems, where a muon orbits a proton or a nucleus, are key tools for precision measurements relevant to atomic and nuclear physics.

Traditionally, the size of a proton was measured either with electron scattering experiments or with atomic spectroscopy. The CODATA 2010 evaluation, compiling data from both these sources, provided a value of the proton charge radius of $r_p = 0.8775(51)$ fm [1]. In contrast, the Charge Radius Experiment with Muonic Atoms (CREMA) collaboration measured the proton radius via laser spectroscopy of the Lamb shift [2] — the 2S-2P$_{\frac{1}{2}}$ atomic transition — in muonic hydrogen ($\mu$-H) — a system in which a muon orbits a proton. The first results were published in 2010 [3] and later confirmed in 2013 [4]. The proton radius was found to be $r_p = 0.84087(39)$ fm [4], an order of magnitude more precise, but surprisingly in disagreement with the accepted CODATA value. Subsequently the CODATA 2014 compilation updated their proton radius value to $r_p = 0.8751(61)$ fm, holding still a substantial disagreement of 5.6 standard deviations ($\sigma$) [5].

Seeking an explanation to the proton radius puzzle, different interpretations of the discrepancy have been suggested, such as systematic re-examinations of electron scattering data [6,7], novel aspects of hadron structure [8,9] and beyond standard-model theories leading to lepton universality violation, see, e.g., the review of Ref. [10]. New experiments were performed or are being performed. These account for precise measurements of electron-proton at low momentum transfer, e.g., muon-proton scattering experiment (MUSE) being commissioned at PSI [11] and the Proton Radius (PRad) experiment at JLab [12], that recently measured a small radius, consistent with the muonic atom results. Furthermore, new electron scattering investigations at low-momentum transfer where obtained using the initial state radiation (ISR) method [13] in Mainz, but unfortunately they suffer from rather large uncertainties. Interestingly, three new spectroscopy measurements in regular Hydrogen have recently been published. The 2S-4P measurement from Garching [14] and the Lamb shift measurement from York [15] obtain a small radius in agreement with the muonic hydrogen results, while the Paris measurement of the 1S-3S transition [16] extracted a large radius. The present situation with all the above mentioned results is depicted in Figure 1.

On a different front, the CREMA collaboration aims at extracting charge radii from Lamb shift measurements on other light muonic atoms, to see whether disagreements persist or not in systems with a different number of protons or neutrons [19]. Recent laser spectroscopy experiments in muonic deuterium ($\mu$-$^2H$) led to the discovery of the "deuteron radius puzzle" [20], which is rather similar to, but not independent from, the proton radius puzzle. Results on Helium isotopes will be released in the near future and laser spectroscopy experiments on muonic Lithium and Beryllium are being planned [21]. For these experiments to be successful, accurate theoretical information on the nuclear structure corrections to the Lamb shift is needed. This motivates the work of this paper.

In Lamb shift experiments the charge radius is extracted from the following equation (in unit of $\hbar = c = 1$) [22]

$$\delta_{\mathrm{LS}} = \delta_{\mathrm{QED}} + \mathcal{A}_{\mathrm{OPE}} r_c^2 + \delta_{\mathrm{TPE}}.$$

While $\delta_{\mathrm{LS}}$ is the measured Lamb shift and $r_c^2$ is the radius one wants to extract, the other terms must be provided by theory. The first term, $\delta_{\mathrm{QED}}$, accounts for quantum electrodynamic corrections, while the other two terms are nuclear structure corrections. The term $\mathcal{A}_{\mathrm{OPE}} r_c^2$ enters at order $(Z\alpha)^4$ and is the energy shift resulting from the finite size of the nucleus. The second term, $\delta_{\mathrm{TPE}}$ — arising from order $(Z\alpha)^5$ — is the energy shift resulting from the two-

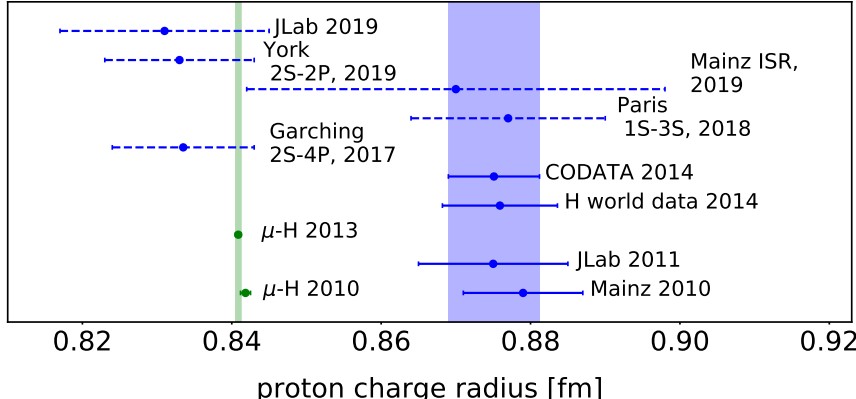

Figure 1: Compilations of proton charge radius determinations. Most recent results are shown as dashed lines, green results stand for muonic spectroscopy while blue results (dashed and continuous lines) are from experiments with the electron-proton system: PRad at JLab 2019 [12], York [15], Mainz ISR [13], Paris [16], Garching [14], CODATA 2014 and H world data [5], $\mu$-H 2013 [4], $\mu$-H 2010 [3], and electron scattering data from JLab [17] and Mainz [18]. Colored bands indicate the uncertainty of the CODATA 2014 and $\mu$-H 2013 data to guide the eye towards the original 5.6 $\sigma$ puzzle. See text for details.

photon exchange interaction. Although this last term accounts for the smallest correction to the Lamb shift, of the order of one percent, it is the largest source of uncertainties. The uncertainty related to this terms limits the precision of the charge radius extraction from laser spectroscopy in light muonic atoms. Nuclear structure corrections have been studied by various groups, see, e.g., Refs. [23–26].

## 2 Comparison of uncertainties

To evaluate the energy corrections due to the two-photon exchange (TPE) diagram, one needs information on the electromagnetic excitation of the nucleus. In the early times this was provided either by photo-absorption cross section data [27,28] or by theoretical calculations with simple models [29]. These approaches however lack accuracy. Theoretical calculation using state-of-art nuclear potentials [22,23,30–33] have significantly improved the accuracy, in some cases also by a factor of 3. To appreciate this, in Table 1 we compare the relative uncertainties in the TPE term obtained with state-of-art nuclear potentials to previous TPE estimates, and display them against to the experimental precision accessible by laser spectroscopy in muonic atoms.

In muonic Lithium atoms — systems where a muon orbits a $^{6,7}$Li nucleus — only estimates based on experimental data are available for the TPE correction. These are plagued by large uncertainties which makes it impossible to get the best out of the experimental precision. Given that few-body calculations have succeeded in obtaining a sizable reduction of the uncertainties, we expect that also in the case of Lithium atoms calculations using state-of-art nuclear potentials will be able to reduce the uncertainty. Here, we set the first steps towards

Table 1: Uncertainty bars in meV of the experimental Lamb shift measurements (Exp), in comparison to the uncertainties of estimates prior to the discovery of the proton-radius puzzle (Estim) and recent few-body calculations with nuclear potentials (Ab initio).

| Atom | Exp | Estim | Ab initio |
|---|---|---|---|
| $\mu^2\text{H}$ | 0.0034 [20] | 0.03 [30] | 0.02 [31] |
| $\mu^3\text{He}^+$ | 0.08 [34] | 1.0 [28] | 0.3 [32] |
| $\mu^4\text{He}^+$ | 0.06 [34] | 0.6 [27] | 0.4 [33] |
| $\mu^6\text{Li}^{2+}$ | 0.7 [35] | 4 [28] | // |
| $\mu^7\text{Li}^{2+}$ | 0.7 [35] | 4 [28] | // |

this goal.

## 3  TPE Corrections

The TPE contribution in muonic atoms contains corrections from the $A$-nucleon dynamics and the intrinsic nucleon structure term $\delta^{N1}$. The $A$-nucleons part — the subject of this work — is further separated into an elastic component (Zemach contribution) and an inelastic part (polarizability), so that one obtains

$$\delta_{\text{TPE}} = \delta^A_{\text{Zem}} + \delta^A_{\text{pol}} + \delta^{N}. \tag{1}$$

The Zemach and polarizability corrections are usually separated into different contributions

$$\delta^A_{\text{pol}} \quad = \quad \delta^{(0)}_{\text{D1}} + \delta^{(0)}_{\text{C}} + \delta^{(0)}_{\text{L}} + \delta^{(0)}_{\text{T}} + \delta^{(0)}_{\text{M}} + \delta^{(1)}_{\text{Z1}} + \delta^{(1)}_{\text{Z3}} + \tag{2}$$

$$+ \quad \delta^{(1)}_{\text{R1}} + \delta^{(1)}_{\text{R3}} + \delta^{(2)}_{\text{NS}} + \delta^{(2)}_{\text{R}^2} + \delta^{(2)}_{\text{Q}} + \delta^{(2)}_{\text{D1D3}}, \tag{3}$$

$$\delta^A_{\text{Zem}} \quad = \quad -\delta^{(1)}_{\text{Z3}} - \delta^{(1)}_{\text{Z1}}. \tag{4}$$

The above terms are all of order $(Z\alpha)^5$, but the Coulomb term $\left(\delta^{(0)}_{\text{C}}\right)$ which is logarithmically enhanced to $(Z\alpha)^6 \log(Z\alpha)$. We include it in our $\delta_{\text{TPE}}$, consistently with Ref [22], where it is also possible to find a full compilation and derivation of these expressions. The numerical superscript stands for the order of an expansion over a parameter $\eta \sim \sqrt{m_r/m_p} \simeq 0.33$. In this formalism the expansion is necessary for obtaining closed forms of the energy corrections. Recently a new formalism has been developed which makes it possible to compute the TPE energy corrections without this expansion [36], but it has so far only be applied to muonic deuterium.

Part of the leading order contributions $\left(\delta^{(0)}_{\text{D1}} + \delta^{(0)}_{\text{C}} + \delta^{(0)}_{\text{L}} + \delta^{(0)}_{\text{T}}\right)$ are expressed in terms of

---

[1]Expressions of $\delta^{N}$ can be found in Eq. (3a), (105) and (106) of Ref. [22].

the dipole response function $S_{\mathrm{D1}}(\omega)$ as

$$\delta_{\mathrm{D1}}^{(0)} = -\frac{16\pi^2}{9}(Z\alpha)^2\phi^2(0)\int_0^\infty d\omega\sqrt{\frac{2m_r}{\omega}}S_{\mathrm{D1}}(\omega), \tag{5}$$

$$\delta_{\mathrm{C}}^{(0)} = -\frac{16\pi^2}{9}(Z\alpha)^3\phi^2(0)\int_0^\infty d\omega\frac{m_r}{\omega}\ln\frac{2(Z\alpha)^2m_r}{\omega}S_{\mathrm{D1}}(\omega), \tag{6}$$

$$\delta_{\mathrm{L}}^{(0)} = \frac{32\pi}{9}(Z\alpha)^2\phi^2(0)\int_0^\infty d\omega\left[\mathcal{F}_{\mathrm{L}}(\omega/m_r)+\frac{\pi}{2}\sqrt{\frac{2m_r}{\omega}}\right]S_{\mathrm{D1}}(\omega), \tag{7}$$

$$\delta_{\mathrm{T}}^{(0)} = \frac{16\pi}{9}(Z\alpha)^2\phi^2(0)\int_0^\infty d\omega\mathcal{F}_{\mathrm{T}}(\omega/m_r)S_{\mathrm{D1}}(\omega), \tag{8}$$

where $m_r$ is the nucleus-muon reduced mass and $\phi^2(0)=(m_rZ\alpha)^3/8\pi$ is the squared muonic 2S-state wave function. The functions $\mathcal{F}_{\mathrm{L/T}}$ are defined as

$$\mathcal{F}_{\mathrm{L}}(\omega/m_r) = \sqrt{\frac{\omega-2m_r}{\omega}}\operatorname{arctanh}\sqrt{\frac{\omega-2m_r}{\omega}}-\sqrt{\frac{\omega+2m_r}{\omega}}\operatorname{arctanh}\sqrt{\frac{\omega}{\omega+2m_r}} \tag{9}$$

$$\mathcal{F}_{\mathrm{T}}(\omega/m_r) = \frac{\omega}{m_r}+\frac{\omega}{m_r}\ln2\frac{\omega}{m_r}+\left(\frac{\omega}{m_r}\right)^2\mathcal{F}_{\mathrm{L}}(\omega/m_r), \tag{10}$$

respectively. The dipole response function $S_{\mathrm{D1}}(\omega)$ can be related to the photo-absorption cross section $\sigma_\gamma(\omega)$ using the following relation

$$S_{\mathrm{D1}}(\omega)=\frac{9}{16\pi^3\alpha\omega Z^2}\sigma_\gamma(\omega). \tag{11}$$

The next-to-leading order Zemach terms $\left(\delta_{\mathrm{Z1}}^{(1)},\ \delta_{\mathrm{Z3}}^{(1)}\right)$ can be computed from the proton $\rho_0^{\mathrm{p}}(\mathbf{R})$ and neutron $\rho_0^{\mathrm{n}}(\mathbf{R})$ ground state density functions as

$$\delta_{\mathrm{Z1}}^{(1)} = 8\pi m_r(Z\alpha)^2\phi^2(0)\int_0^\infty\int_0^\infty d^3Rd^3R'|\mathbf{R}-\mathbf{R}'|\rho_0^{\mathrm{p}}(\mathbf{R})\left[\frac{2}{\beta^2}\rho_0^{\mathrm{p}}(\mathbf{R}')-\lambda\rho_0^{\mathrm{n}}(\mathbf{R}')\right], \tag{12}$$

$$\delta_{\mathrm{Z3}}^{(1)} = \frac{\pi}{3}m_r(Z\alpha)^2\phi^2(0)\int_0^\infty\int_0^\infty d^3Rd^3R'|\mathbf{R}-\mathbf{R}'|^3\rho_0^{\mathrm{p}}(\mathbf{R})\rho_0^{\mathrm{p}}(\mathbf{R}'). \tag{13}$$

Finally, one of the next-to-next-to-leading order term is obtained as

$$\delta_{\mathrm{NS}}^{(2)}=-\frac{128\pi^2m_r^2}{9}(Z\alpha)^2\phi^2(0)\left[\frac{2}{\beta^2}+\lambda\right]\int_0^\infty d\omega\sqrt{\frac{\omega}{2m_r}}S_{\mathrm{D1}}(\omega), \tag{14}$$

where $\beta=\sqrt{12/r_{\mathrm{p}}^2}$ and $\lambda=-r_{\mathrm{n}}^2/6$ with $r_{\mathrm{n}}$, $r_{\mathrm{p}}$ denoting the neutron and proton charge radius. In essence, each of these energy corrections involves either an integration of nuclear electromagnetic response functions over energy or an integration of proton/neutron densities over distance.

## 4 Results

In this work we set the first steps towards an ab initio computation of $\delta_{\mathrm{TPE}}$ for muonic Lithium atoms. For the terms related to $S_{\mathrm{D1}}(\omega)$ we start from photo-absorption cross sections calculated for $^6\mathrm{Li}$ and $^7\mathrm{Li}$ in Refs. [37–39] using hyperspherical harmonics expansions with the AV4' [40] potential.

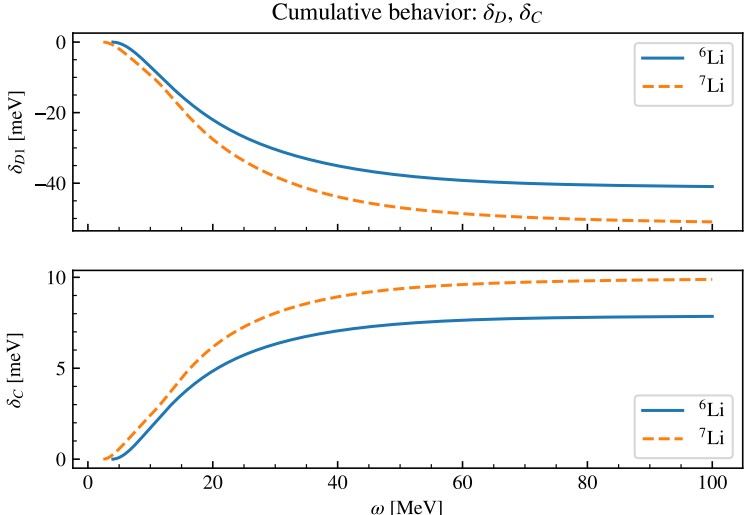

Figure 2: The $\delta_{\mathrm{D1}}^{(0)}$ and $\delta_{\mathrm{C}}^{(0)}$ contributions to the Lamb shift in $\mu$-$^6$Li$^{2+}$ and $\mu$-$^7$Li$^{2+}$ atoms as a function of the energy cut-off in the photo-absorption cross section. The latter has been computed using the AV4' interaction.

In Figure 2 we show our analysis of the $\delta_{\mathrm{D1}}^{(0)}$ and $\delta_{\mathrm{C}}^{(0)}$ contributions to $\delta_{\mathrm{TPE}}$ in $\mu$-$^6$Li$^{2+}$ and $\mu$-$^7$Li$^{2+}$ atoms. The photo-absorption cross section $\sigma_\gamma(\omega)$, and consequently $S_{\mathrm{D1}}(\omega)$, have been calculated only for energies up to 100 MeV. At this energy, the integrals determining the nuclear structure corrections might not yet be fully converged. We estimated the truncation errors due to the cut-off in the integration upper limit by taking half of the relative difference between computations with cuts at 80 MeV and 100 MeV, respectively. From Figure 2 it is clear that for $\delta_{\mathrm{D1}}^{(0)}$ and $\delta_{\mathrm{C}}^{(0)}$ convergence in the upper integration limit has been reached and accordingly the uncertainties are small compared to the strength of the corrections. However we found that, for $\delta_{\mathrm{L}}^{(0)}$, $\delta_{\mathrm{T}}^{(0)}$ and $\delta_{\mathrm{NS}}^{(2)}$, the convergence is slower, with $\delta_{\mathrm{NS}}^{(2)}$ being the slowest. Although this is taken into account by the larger relative uncertainties, a further analysis extending the calculation of the photo-absorption cross sections to higher energies will be performed in the future.

In order to compute the Zemach terms $\left(\delta_{\mathrm{Z1}}^{(1)}, \delta_{\mathrm{Z3}}^{(1)}\right)$ we made use of densities calculated with variational Monte Carlo algorithms, which used the AV18+UX potential [41, 42]. The computational procedure involves an interpolation over the density-data points followed by a numerical integration of Eq. (12) and Eq. (13). The densities are provided up to large distance, so that the convergence of these integrals is not an issue. However, given that the data points have a statistical uncertainty, to estimate how these uncertainties propagates into the Zemach corrections we made use of a Monte Carlo statistical simulation. We generated new density-data points following the original distributions — with every point subject to a Gaussian distribution with mean and standard deviations corresponding to the central value and standard deviations of each point of Ref. [41]. For every simulation we interpolated and computed the relative integral, maintaining the normalization of the density. We thus obtain a distribution of $\delta_{\mathrm{Z1}}^{(1)}$ and $\delta_{\mathrm{Z3}}^{(1)}$ from which we extracted mean and standard deviation, the latter being the statistical uncertainty. Figure 3 shows the statistical distribution of the so obtained $\delta_{\mathrm{Z1}}^{(1)}$ and $\delta_{\mathrm{Z3}}^{(1)}$ correction in $\mu$-$^6$Li$^{2+}$. Similar plots are obtained for $\mu$-$^7$Li$^{2+}$. From Eq. (1) it is clear that $\delta_{\mathrm{Z1}}^{(1)}$ and $\delta_{\mathrm{Z3}}^{(1)}$ cancel out when considering the total TPE correction, however we still compute them with the purpose of comparing to Ref. [28], who has estimated only the

inelastic part of $\delta_{\text{TPE}}$.

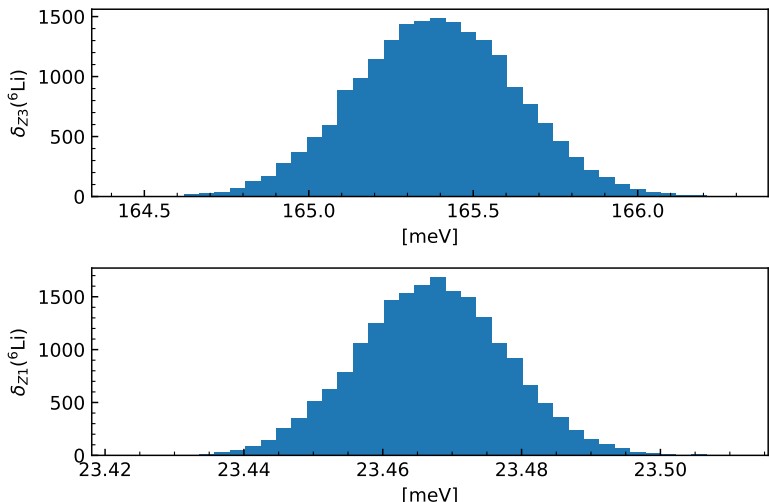

Figure 3: Distribution of the $\delta_{Z1}^{(1)}$ and $\delta_{Z3}^{(1)}$ for $^6$Li obtained from the Monte-Carlo analysis.

We summarize these results showing a compilation of the computed terms in Table 2, where we show the uncertainty associated to each term computed as explained above.

Table 2: Nuclear structure corrections to $\mu$-$^6$Li$^{2+}$ and $\mu$-$^7$Li$^{2+}$ atoms. The uncertainties reported for $\delta_{Z1}^{(1)}$ and $\delta_{Z3}^{(1)}$ are statistical and are obtained with a Monte Carlo analysis, while the uncertainties in all the other dipole-like terms are systematic and due to the truncation on the upper limit of the energy integration.

| Atoms | $\delta_{Z1}^{(1)}$ | $\delta_{Z3}^{(1)}$ | $\delta_{D1}^{(0)}$ | $\delta_{C}^{(0)}$ | $\delta_{L}^{(0)}$ | $\delta_{T}^{(0)}$ | $\delta_{NS}^{(2)}$ |
|---|---|---|---|---|---|---|---|
| $\mu$-$^6$Li$^{2+}$ | 23.47(1) | 165.4(2) | $-41.0(2)$ | 7.85(3) | 1.66(3) | $-0.75(1)$ | $-1.41(3)$ |
| $\mu$-$^7$Li$^{2+}$ | 22.03(1) | 126.5(2) | $-51.0(3)$ | 9.89(4) | 2.04(4) | $-0.92(2)$ | $-1.75(4)$ |

The terms $\delta_{R1}^{(1)}$ and $\delta_{R3}^{(1)}$ cannot be computed yet, as one needs the off diagonal proton-proton density distribution, which is not available in Ref. [41]. They are expected to be large and with absolute values of the same order of $\delta_{Z1}^{(1)}$ and $\delta_{Z3}^{(1)}$. With the goal of comparing our numbers with previous studies by Drake et al. [28], we thus estimate these terms assuming the ratio $\delta_{Z1}^{(1)}/\delta_{R1}^{(1)}$ and $\delta_{Z3}^{(1)}/\delta_{R3}^{(1)}$ behave as observed in $\mu$-$^3$He$^+$ and $\mu$-$^4$He$^+$.

In Figure 4 we show all the terms together, the calculated ones and the estimated ones, in a graphical way. Even though we still miss a few terms to compose the total TPE, we estimate their effect to be only at the level of a few percent, based on observations made on other muonic atoms. When we sum all our terms we obtain preliminary values which are of the same order of magnitude as the estimates provided by Drake et al. In particular, for $\mu$-$^6$Li$^{2+}$ while Ref. [28] quoted $-15\pm4$ meV we get $-11.8$ meV with a lower bound uncertainty of 0.3 meV, whereas for $\mu$-$^7$Li$^{2+}$ Ref. [28] obtained $-21\pm4$, while we get $-22.2$ meV, with a lower bound uncertainty of 0.4 meV. The given uncertainties are quadrature sums of the uncertainties reported in Table 2 for each term. We stress that this lower bound uncertainty is coming only from the numerical source of error and all the other uncertainty sources still need to be studied. We expect the

potential model dependences to be the largest source of error. Further investigation is needed to include all missing terms and to assess an overall solid error bar.

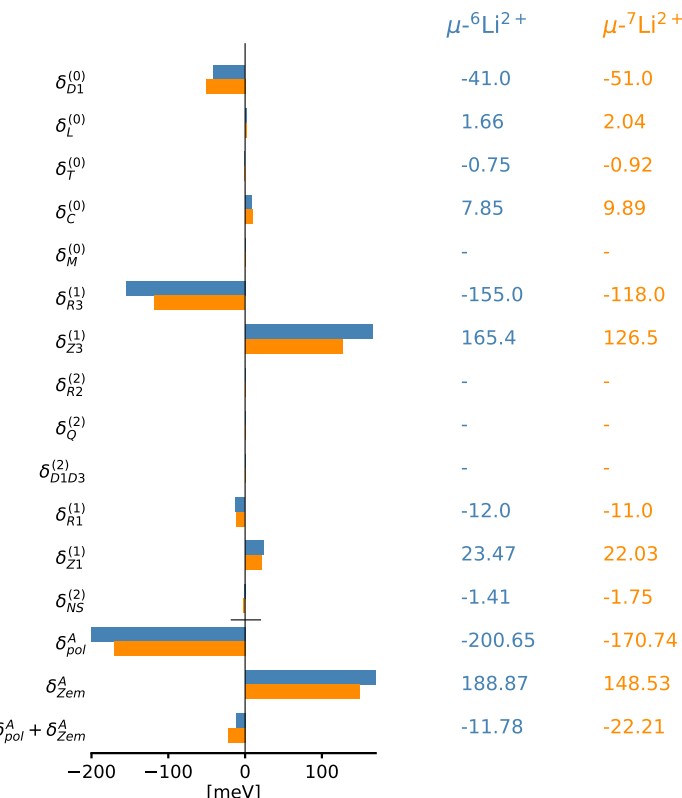

Figure 4: Graphical representation of preliminary nuclear structure corrections to $\mu$-$^6$Li$^{2+}$ and $\mu$-$^7$Li$^{2+}$ atoms in meV.

## 5  Conclusion

We computed the dipole and Zemach terms contributing to the TPE correction to muonic Lithium atoms. Dipole terms are obtained starting from an ab initio photo-absorption calculation performed with the AV4' potential. The Zemach terms are obtained from an integration of the variational Monte Carlo computation of the $^{6,7}$Li$^{2+}$ charge density distributions using the AV18+UX interaction. While the calculations are obtained with different interactions, our results constitute the first steps towards a microscopic computation of the TPE corrections in muonic Lithium atoms. In particular, we are missing a computation of the monopole, quadrupole, magnetic-dipole and the $D_1D_3$ response functions, as well as a rigorous evaluation of the important $\delta_{R3}^{(1)}$ and $\delta_{R1}^{(1)}$ corrections [22]. To compare with previous results, we estimate these latter terms from the scaling observed in other muonic atoms, and show that our results are consistent with previous literature.

Future work will be devoted to a complete and consistent evaluation of these terms using

realistic nucleon-nucleon and three body forces.

## Acknowledgements

**Funding information**  This work was supported by the Deutsche Forschungsgemeinschaft (DFG) through the Collaborative Research Center [The Low-Energy Frontier of the Standard Model (SFB 1044)], and through the Cluster of Excellence "Precision Physics, Fundamental Interactions, and Structure of Matter" (PRISMA$^+$ EXC 2118/1) funded by the DFG within the German Excellence Strategy (Project ID 39083149).

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
