# Peer review of "Muonic Lithium atoms: nuclear structure corrections to the Lamb shift"

_SciPost Physics Proceedings, doi:SciPost Phys. Proc. 3, 028 (2020)_

## Round 1 · Referee Report · Krzysztof Pachucki · 2019-12-3

Report

This is a valuable work. The calculation of the leading nuclear
structure corrections to the Lamb shift in muonic lithium atoms
is crucial for the interpretation of the planned measurements.
However, several specific comments or questions shall be addressed
before approving this work for the publication, namely:

Requested changes

1: the proton radius puzzle is merely the disagreement between
Paris measurements in hydrogen with all the rest.
CODATA 2014, H world data 2014 are all based on Paris measurements,
so they are not independent, as Figure 1 would suggest.
Moreover, the recent PRAD value (published in NATURE)
of the proton charge radius is also in agreement with muH rp.
CODATA 2018 also assumed smaller proton radius.
So to my opinion, there is no basis for heating this proton radius puzzle.

2: Authors should clarify the meaning of Lamb shift in \delta_{LS},
it is not 2S-2P but 2S - 2P_{1/2} energy difference.

3: It is not clear what Authors mean by discovery of the deuteron radius puzzle,
since r_d^2 - r_p^2 agrees between electronic and muonic measurements.
Do Authors consider it to be an independent puzzle ?

4: Authors define \delta_{TPE} as a nuclear structure corrections of order (Z alpha)^5,
however \delta_C^{(0)} in Eq. (6) is of higher order in Z alpha.
What is the basis to call it a two-photon exchange TPE correction ?

5: \delta^N in Eq. (1) is not defined anywhere.

6: After Eq. (8) Authors claim the \phi^2(0) is a squared wave function,
define \beta and \lambda which have not yet been used so far. Moreover,
if r_n is defined to be the neutron charge radius, how come r_n^2 is negative.

7: Authors elaborate on page 6 the calculations of \delta_{Z1}^{(1)} and \delta_{Z3}^{(1)}
which obviously cancel out in the sum in Eq. (1), explanations are needed.

8: A better representation of works by others on nuclear structure effects would be in place.

---

## Round 2 · List of Changes

We thank the referee for his valuable comments. We would like to address his points below.

1) In our first paragraphs we describe the history of the proton radius puzzle, not only what it is
today. We do not feel that stating that the proton radius puzzle is only a disagreement of the
Paris measurement with the rest is fair, because there are electron scattering experiments from
Mainz that measured a large radius. As nuclear physicists we want to mention them and we
think it is important to understand why they give a large radius. as opposed to the small radius
found by PRAD.
The PRAD Nature paper was published on Nov 6th, while we submitted the proceedings on
Nov 1st. That is why we did not include it in the figure. Of course, we were aware of these data
(as preliminary data) but we did not want to include unpublished data, especially when they are
from other collaborations. Now that the paper is published, we added the PRAD point to Fig. 1.
We also modified the paragraph:
"New experiments are also being performed. These account for precise measurements of
electron-proton at low momentum transfer, e.g., the Proton Radius (PRad) experiment at JLab
[11] and the muon-proton scattering experiment (MUSE) being commissioned at PSI [12]“
to
"New experiments were performed or are being performed. These account for precise
measurements of electron-proton at low momentum transfer, e.g., muon-proton scattering
experiment (MUSE) being commissioned at PSI [11] and the Proton Radius (PRad) experiment
at JLab [12], that recently measured a small radius, consistent with the muonic atom results.
Furthermore, new ...”

2) We corrected to 2S-2P_1/2.

3) We do not consider the deuteron radius puzzle an independent puzzle, but it has been
coined "deuteron radius puzzle" in Ref.[20], so we think we can call it this way. However, to
take into account the referee's point we substituted the sentence:
" Recent laser spectroscopy experiments in muonic deuterium (μ-2 H ) led to the discovery of
the "deuteron radius puzzle" [20], which is rather similar to the proton radius puzzle."
=>
" Recent laser spectroscopy experiments in muonic deuterium (μ-2 H ) led to the discovery of
the "deuteron radius puzzle" [20], which is rather similar to, but not independent from, the
proton radius puzzle.

4) We want to be consistent with the notation of the our recent JPG review (Ref.[22]), so we call
it \delta_TPE. Of course the Coulomb term is not of order (Z\alpha)^5, and the referee is correct
in saying that it is not clearly explained. To amend, right after Eq.(4), we have added the
following sentence:
“ The above terms are all of order $(Z\alpha)^5$, but the Coulomb term (\delta_C^{(0)}) which is
logarithmically enhanced to $(Z\alpha)^6\log(Z\alpha$). We include it in our $\delta_{\text{TPE}}$,
consistently with Ref. [22], where it is also possible to find a full compilation and derivation of these
expressions.”

5) We did not include the expression of \delta_N because it is not the main point of the
proceedings. However, to be more clear we added a footnote:\footnote{Expressions of \delta_N can be found in Eq. (3a),(105) and (106) of Ref.[22].}

6) We corrected for that by moving the definitions of \beta and \lambda after Eq. (14). The squared
neutron charge radius is negative, even though the neutron is neutral, because it can be pictured as
a proton with a negative pion cloud that extends further at larger distance, so that the second
moment of the charge distribution is slightly negative.

7) We added the explanation:
"From Eq.(1) it is clear that $\delta_{\text{Z1}}^{(1)}$ and $\delta_{\text{Z3}}^{(1)}$ cancel out
when
considering the total TPE correction, however we still compute them with the purpose of
comparing to Ref. [28], who has estimated only the inelastic part of $\delta_{\text{TPE}}$."

8) We have added the following sentence with 4 new citations, before Section 2:
“Nuclear structure corrections have been studied by various groups, see, e.g., Refs. [23-26].”

We have performed one other change:
* Corrected the value of \eta to 0.33

We hope that now the manuscript can be accepted for publication as a proceedings of the few-
body conference 2019 in Surrey.

With best regards,
Simone, Anna and Sonia

---

## Editorial Decision

published